# Time Series Analysis of Forest Dynamics at the Ecoregion Level

**Olga Rumyantseva [1,\*], Andrey Sarantsev [2] and Nikolay Strigul [1]** 

1   Department of Mathematics and Statistics, Washington State University Vancouver, 14204 NE Salmon Creek Avenue, Vancouver, WA 98686, USA; nick.strigul@wsu.edu

2   Department of Mathematics and Statistics, University of Nevada in Reno, Reno, NV 89557, USA; asarantsev@unr.edu

\*   Correspondence: olga.rumyantseva@wsu.edu

**Abstract:** Forecasting of forest dynamics at a large scale is essential for land use management, global climate change and biogeochemistry modeling. We develop time series models of the forest dynamics in the conterminous United States based on forest inventory data collected by the US Forest Service over several decades. We fulfilled autoregressive analysis of the basal forest area at the level of US ecological regions. In each USA ecological region, we modeled basal area dynamics on individual forest inventory pots and performed analysis of its yearly averages. The last task involved Bayesian techniques to treat irregular data. In the absolute majority of ecological regions, basal area yearly averages behave as geometric random walk with normal increments. In California Coastal Province, geometric random walk with normal increments adequately describes dynamics of both basal area yearly averages and basal area on individual forest plots. Regarding all the rest of the USA's ecological regions, basal areas on individual forest patches behave as random walks with heavy tails. The Bayesian approach allowed us to evaluate forest growth rate within each USA ecological region. We have also implemented time series ARIMA models for annual averages basal area in every USA ecological region. The developed models account for stochastic effects of environmental disturbances and allow one to forecast forest dynamics.

**Keywords:** time series forecasting; North American ecoregions; forest dynamics; autoregressive models; random walk model; AR(1) process; ARIMA

## 1. Introduction

### 1.1. Background

Forecasting of forest dynamics at a large scale is essential for land use management, climate change predictions and global biogeochemistry modeling. Forest modeling is challenging due to the ecological complexity of a forest as a complex adaptive system [1]. Forested ecosystems demonstrate self-organization at different spatial and temporal scales. Current modeling approaches such as the Markov chain model [2–5], individual-based models [6–11] and differential equations models [12] are powerful predictive tools, especially when applied to forest dynamics at local and intermediate spatial scales; however, they have particular limitations in capturing continuous large scale forest dynamics. These challenges include capturing forest changes under non-stationary disturbance regimes caused by climatic changes and changes in land-use practices. Changes in frequencies of primary natural disturbances such as forest fires, droughts and hurricanes occur at random, yet they non-linearly depend on climatic changes. Markov chain models [13–15] can successfully describe large scale forest dynamics; however, this approach is based on breaking continuous biomass into discrete

bins [13]. Spatially-explicit individual-based models [7,9,11] simulate forest dynamics in continuous time and agregated biomass characteristics are also continuous variables; however, these models are not analytically tractable and require great computer power to be applied at large spatial scales. Time series modeling traditionally employed in financial mathematics, meteorology and remote sensing is a promising tool for the forecasting of forest dynamics at large scales. We recently demonstrated the utility of autoregressive models for modeling forest dynamics in Quebec [16]. Time series models operate in continuous time and with continuous state space [16]. Similarly to Markov chains, time series models can be parameterized from irregularly collected forest inventory data using Bayesian techniques [14–16]. Autoregressive models are computationally efficient and analytically tractable, and, therefore, are especially suitable for large scale carbon cycle and biogeochemistry forecasting models, which do not require species-level predictions of forest dynamics [17].

### 1.2. Forecasting of Forest Dynamics: From Individual Trees to Ecoregions

Forests have complex hierarchical organizations, and different levels of said hierarchy have been accounted for while building models. The biocomplexity challenges for robust forest forecasting are to understand what forces drive forest community structure and dynamics (and how they do so). Certain limitations of traditional experimental approaches make forecasting modeling especially important. In particular, experimental studies often concentrate on one focal level of organization, while ignoring processes at the other scales. Conclusive experimental results to support land-management decisions to utilize different silvicultural techniques may not be obtained on a reasonable time scale and can be too expensive. Forest yield tables are one of the oldest biological models, with more than a 200-year history of development and practical applications for tree plantations with reduced tree competition and homogeneous spatial patterns [18,19]. However, forest yield tables have an empirical nature and limited applications for more spatially heterogeneous silvicultural systems with intensive crown competition.

Individual-based forest models (IBMs) [7,8,20,21] consider birth, growth and mortality of certain tree species competing for resources [11,22]. These models, similarly to the growth-tables, regard the stand-level of forest hierarchical organization, which comes after the individual-level. Forest growth IBMs developed in two distinct ways. Foresters have developed stand simulators in order to estimate and optimize stand production; meanwhile, ecologists sought to study basic processes, such as the succession, coexistence and dynamics of native forests. This difference in initial goals is reflected in the model structures, as forester and ecological models each concentrate on different aspects of forest development. Ecological models, such as the family of gap models originated from JABOVA [6,19,23] typically include detailed descriptions of the targeted processes that are considered to be most important; for example, JABOVA is focused on succession and gap dynamics [6]. Forester IBFMs, such as TASS [18], usually target plantations, and focus on overstory dynamics and detailed descriptions of individual tree growth in the given neighborhoods, ignoring seed production, gap and understory dynamics. Moreover, in ecological models the functions describing tree growth are generally selected to scale up and approximate the smaller scale physiological processes inside an individual tree; meanwhile, forester simulators often employ empirical regression functions based on yield tables, without consideration of the underlying mechanisms. In particular, forester stand simulators are traditionally individually-based, where tree competition for light is described by competition indices. While the empirical approach developed in the practical stand simulators does not lead to a better understanding of ecological phenomena such as functional trade-offs, these models do yield very good predictions of stand growth, especially in plantations [24]. This in part is because the functions employed to approximate forest yield tables can be easily parameterized using large sets of available empirical data; and in part is because plantations exhibit less species diversity and more regular spatial patterns than do native forests. Both forester and ecological models such as TASS [18], SILVA [25], BALANCE [26] and SORTIE [8], and other gap models [7], share a similar assumption concerning tree growth: i.e., trees are assumed to grow vertically; therefore, the zone

of influence is centered at the stem base. As a result, these models do not introduce tree leaning as a mechanism of adaptive tree-morphological plasticity. This simplification is more justified for forester models, because the typical silvicultural methods such as thinning regimes lead to the spatially homogeneous stands and limit crown competition. Ecological individual-based forest simulators embody a wide range of simplifying assumptions with respect to crown structure, which were addressed in new generation models focused on crown plasticity and spatial competition [11,12,27,28]. Several approaches were proposed for scaling forest stand dynamics from the stand level to large spatial scales. The forest gap models, or their approximations, such as the ecosystem demography model [10] and the perfect plasticity approximation model [11], can be scaled up directly to the large scales using spatially-explicit computer simulations of forest stands and large spatial units [29,30]. This approach allows one to link forest dynamics with biogechemistry models [17,31]. Analytically tractable approaches for scaling up mosaic of forest patches include continuous and discrete conservation law models. JABOVA-FORET models [6] and many of their descendants, gap models, are based on the premise that a forest can be represented as a mosaic of homogeneous patches, i.e., gaps, each of which can be modeled independently. The size of every gap is usually assumed to be equal to the size of one large overstory tree. These patches have a horizontally homogeneous structure—i.e., the crowns of all trees in a gap extend horizontally over each patch. A forest patch mosaic can be extended to the large spatial areas using continuous conservation law models represented by partial differential equations [22,32]. The discrete conservation law models are typically represented by Markov chains [5,12].

Autoregressive models developed in this paper operate with stand-level forest patch dynamics at the ecoregion level. We model aggregated dynamics of individual trees growing on a certain forest patch (i.e., a forest stand represented in the dataset as one permanent forest inventory plot). The ecoregions are areas characterized by similar environmental conditions, including climatic factors, disturbance regimes, soil types and vegetation. In this work we employ the US Forest Service ecoregion classification developed by R.G. Bailey [33,34]. This classification divides the conterminous United States into biogeographic domains, which are divided into smaller biogeographic divisions. Finally, the biogeographic divisions are divided into biogegraphic provinces [33]. In this work we applied the forest patch dynamic conceptual framework to ecoregions that are biogeographic provinces (see Appendix A). The Bailey's provinces were defined using climatic and biogeographic criteria which did not include patterns of vegetation dynamics [33,34]. Recently, this ecoregion classification was linked to the forest succession mechanisms driven by the shade-tolerance and forest gap dynamics [15]. The presented work aimed to compare forest dynamic patterns between ecoregions using time series analysis in order to improve our understanding about the large scale effects of climate and disturbance regimes on the North American forests.

*1.3. Our Contributions*

In the present paper, we extend the autoregressive modeling approach recently developed to forecast forest dynamics in Quebec [16] to the conterminous United States (see also Sections 2.2 and 2.3.1), we also examine autoregressive integrated moving average models ARIMA(p,d,q) (see Section 2.4). Within each USA ecoregion, we model forest dynamics using various autoregressive time series models. Forest dynamics are characterized by basal area computed from forest inventory data [13,14]. In autoregressive models, we regress the present values of basal area on the previous ones to obtain predictions of future values.

In the absolute majority of USA ecoregions (Sections 3.2 and 3.3), basal area yearly averages behave as a geometric random walk with normal increments: the present value of basal area is a sum of its previous value, basal area mean and Gaussian white noise (independent and identically distributed random values have zero mean). In California Coastal Province (ecoregions 261 and 263), geometric random walk with normally distributed increments adequately describes the dynamics of both basal area yearly averages and basal area on individual forest plots. Regarding all the rest

USA ecoregions, the autoregressive model describing basal area dynamics on individual forest patches turned out to be random walk with non-normal heavy tails.

The Bayesian approach which we used to treat the absent basal area observations for certain years, allowed us to evaluate the basal area growth rate for each USA location (Section 3.3.2). We obtained the "heat" map of comparable forest growth rate in USA. According to these results, the top three ecological regions showing the most rapid forest growth rate are the following: M262 (California Coastal Range Open Woodland Shrub Coniferous Forest Meadow Province), 342 (Intermountain Semidesert Province), M334 (Black Hills Coniferous Forest Province) and 263 (California Coastal Steppe, Mixed Forest and Redwood Forest Province). Oppositely, the top three ecoregions with the lowest basal area growth rate are: 321 (Chihuahuan Semidesert Province), M313 (Arizona New Mexico Mountains, Semidesert Open Woodland Coniferous Forest Alpine Meadow Province), M341 (Nevada–Utah Mountains Semidesert–Coniferous Forest–Alpine Meadow Province).

We have considered autoregressive integrated moving average models ARIMA(p,d,q) for the description of the ecoregion-level forest dynamics (see Section 2.4 for the definition and technical details). We investigated which simplest ARIMA model better fits basal area annual averages in each ecoregion and obtained the groups of ecoregions based on it. From these results we noticed that the two models ARIMA(0,1,0) (random walk) and ARIMA(0,0,1) (MA(1)) are the best fit in the ecoregions located mainly in Humid USA ecological domain. On the other hand, ARIMA(1,0,0) (AR(1)) and ARIMA(1,1,0) models turned out to be the best models in Dry USA ecological Domain (precisely, ARIMA(1,1,0)—in Mountain provinces of Dry Domain). From this we can conclude that in the Dry Domain the present year basal area values are highly dependent on the previous year's values.

In general, we revealed a strong analogy between the USA's forests and the stock market. Precisely, in the majority of USA ecoregions basal area in individual forest patches behaves as an individual stock: its dynamics can be described by random walk, while its increments are not normally distributed. At the same time, basal area yearly averages, similarly to stock market indices, behave as a geometric random walk with normal increments. Using Bayesian statistical methods, we revealed the diagram with comparable forest growth rate from which we can detect the regions of expanding and shrinking forests. We demonstrated the utility of the Bayesian approach in the estimation of forest dynamics characteristics. Finally, we generalized our research of forest dynamics using autoregressive models AR(1) by moving towards the more general ARIMA models and obtained the diagram of USA locations with the best fitting ARIMA models.

## 2. Materials and Methods

### 2.1. Data Mining of the Forest Inventory and Analysis Data

The present research is based on USDA Forest Inventory and Analysis dataset (FIA data) covering 1968–2013: https://www.fia.fs.fed.us/. The data contain information about 211,949 plots and include 409,868 observations totally. The FIA system of forest inventory plots evenly covers the territory of the conterminous United States to monitor the forest patch mosaic. The inventory plots are observed every few years with irregular time intervals. Such weather and human disturbances as forest fires, droughts, hurricanes and harvesting substantially influence USA forest. For each FIA plot we computed basal area and biomass see [5,13,15] for the details of these calculations. Table 1 contains general information about the investigated data: the list of ecological regions observed, years and correlation coefficients between basal area and biomass calculated all over the USA. The code used for this article is available on GitHub repository Olia8848/US-forest-autoregression.

### 2.2. Autoregressive Model Ar(1) for Basal Area on Individual Forest Plots

The main goal of building autoregressive models for forest basal area is to understand how this characteristic changes in time, and what qualitative laws describe it. Modeling within ecological regions is based on consideration that the tree species belonging to the same ecoregion belong to

the same ecological society of organisms and are subject to the similar disturbances. We built the model described below within each of 36 USA ecoregions.

In each ecoregion, for every forest patch location (i.e., forest inventory plot), we consider the autoregressive model AR(1) (relationship where the next year's basal area is regressed on the previous year's basal area):

$$y_p(t) = r + a \cdot y_p(t-1) + \varepsilon_p(t). \tag{1}$$

In the equation above, $p$ is a certain fixed forest inventory patch (plot); $y_p(t)$ is the logarithm of basal area (we keep logarithmic scale for convenience based on positivity of basal area) at the year $t$; $y_p(t-1)$ is the previous year's logarithm of basal area; constants $r, a$ are the parameters of AR(1) model: $r$—drift (basal area trend), $a$—diffusion; and the variables $\varepsilon_p(t)$ are independent, identically distributed (i.i.d.) random variables with normal distribution $\mathcal{N}(0, \sigma^2)$.

For each patch $p$ we generally have very few observations (on average, each plot was observed only twice). Moreover, for the majority of patches $p$ we do not have records for all consecutive years $t$. Hence, we need some technique to cope with this missing data. This way, we play with the Equation (1) to obtain vectors having a large number of points to perform necessary simulations.

Consider the Equation (1). Let $t$ and $t+s$ be subsequent years when the basal area was measured on the patch $p$. Applying the Equation (1) multiple times to itself, we obtain the equation for subsequent basal area measurements:

$$y_p(t+s) = a^s \cdot y_p(t) + r\left(1 + a + \ldots + a^{s-1}\right) + \sum_{k=1}^{s} \varepsilon_p(t+k) \cdot a^{s-k}. \tag{2}$$

In the Equation (2), the sum of increments have the following distribution:

$$\sum_{k=1}^{s} \varepsilon_p(t+k) \cdot a^{s-k} \sim \mathcal{N}\left[0, \sigma^2(1 + a^2 + \ldots + a^{2(s-1)})\right] \quad \text{i.i.d.} \tag{3}$$

We introduce new notation in order to obtain classical distribution:

$$\sum_{k=1}^{s} \varepsilon_p(t+k) \cdot a^{s-k} = \left[1 + \ldots + a^{2(s-1)}\right]^{1/2} \delta_p(t,s), \tag{4}$$

where the increments $\delta_p(t,s) \sim \mathcal{N}(0, \sigma^2)$ i.i.d. Now, the Equation (2) becomes:

$$y_p(t+s) - a^s \cdot y_p(t) = r\left(1 + \ldots + a^{s-1}\right) + \left[1 + \ldots + a^{2(s-1)}\right]^{1/2} \delta_p(t,s). \tag{5}$$

Dividing the Equation (5) by the sum $Ev(a,s) = \left[1 + a^2 + \ldots + a^{2(s-1)}\right]^{1/2}$ and introducing the notation: $Odd(a,s) = \left[1 + a + \ldots + a^{s-1}\right]$, we obtain the following relationship:

$$[Ev(a,s)]^{-1} \cdot \left[y(t+s) - a^s \cdot y(t)\right] = [Ev(a,s)]^{-1} \cdot Odd(a,s) \cdot r + \delta_p(t,s). \tag{6}$$

Finally, we work with the vector Equation (6). In this equation we can use any pair of basal area observations which happen at times $t$ and $t+s$ at the single patch. This way, for any single forest location, we employ all existing measurements of basal area and use them in the Equation (6).

To parameterize the model, we fix an $a$ value and run the linear regression with respect to $r$. By trying various $a$ values, we choose the ones such that the standard error of the regression (6) is minimal. We build the regression for each $a$ value in the interval $(-2, 2)$ with a step 0.01, and then

we find the *r* value and the standard error $\sigma$. Minimizing standard error for this linear regression is equivalent to maximizing the likelihood function (see [16]).

As a result, in all USA ecoregions the value $a = 1$ gives the minimal standard error for the linear regression (6). The value $a = 1$ makes the model (1) into random walk. However, the residuals of this random walk are not normally distributed in all USA ecoregions except the California Coastal ecoregions 261 and 263, where we have a geometric random walk with normal increments.

### 2.3. Autoregressive Model Ar(1) for Basal Area Yearly Averages

#### 2.3.1. Frequentist Analysis

In the present section, we build a model for basal area yearly averages in each USA ecoregion. Consider first a simple autoregressive random walk model:

$$y(t) = y(t-1) + \varepsilon(t), \quad \varepsilon(t) \sim \mathcal{N}(\mu, \rho^2) \quad \text{i.i.d.}, \tag{7}$$

where $y(t)$ is the mean of basal area logarithms at the year t; $\mu$ the mean of the increments $y(t) - y(t-1)$ and $\rho^2$ is the variance of these increments; and $\varepsilon(t)$ are the model residuals, normally distributed with mean $\mu$ and variance $\rho^2$.

By implementing the model (7) in every ecoregion, we obtained that this model fits (increments are normally distributed) basal area yearly averages in the absolute majority of ecoregions. To evaluate the parameters $\mu$ and $\rho$, we apply the model (7) multiple times to itself and obtain:

$$y(t) = y(0) + \varepsilon(1) + \ldots + \varepsilon(t), \quad \varepsilon(i) \sim \mathcal{N}(\mu, \rho^2) \quad \text{i.i.d.} \tag{8}$$

We rewrite the Equation (8) in the original scale:

$$z(t) = z(0) \exp \left[ \varepsilon(1) + \ldots + \varepsilon(t) \right], \tag{9}$$

where $z(t)$ is the mean of basal area at the year t. From (9), we compute the mean and variance of random variable $z(t)$:

$$\mathrm{E}[z(t)] = z(0) \cdot \exp \left( t(\mu + \sigma^2/2) \right), \tag{10}$$

$$\mathrm{Var}[z(t)] = z^2(0) \cdot \exp(2\mu t + \sigma^2 t) \cdot \left( \exp(\sigma^2 t) - 1 \right). \tag{11}$$

In the model (7), we face a certain difficulty: different years have very distinct numbers of basal area observations. For example, in the ecoregion 251 (Prairie Parkland Province), which was observed during 25 different years, we have 137 observations for the year 1977, 48 observations for the year 1980 and 698 observations for the year 1981. Consequently, we need a more advanced approach to account for existing forest observations while evaluating basal area means and variances. We use the Bayesian approach below to treat uneven number of observations.

#### 2.3.2. Bayesian Analysis

As mentioned in the previous section, different years have different numbers of basal area observations. This led us to use Bayesian statistics to evaluate yearly means and variances of basal area, rather than rely on the empirical (calculated from data) means and variances. We use Bayesian analysis described below in every USA ecological region, as the problem of uneven observations takes place in all 36 ecoregions.

Consider basal area logarithms $x_1(t), x_2(t) \ldots, x_{N_p}(t)$, where $N_p = N_p(t)$ is the number of patches observed in the year $t$. When $N_p$ is a big number (which holds in all 36 ecoregions), we can assume that the basal area logarithms follow normal distribution:

$$x_1(t), x_2(t) \ldots, x_{N_p}(t) \sim \mathcal{N}[m(t), v(t)] i.i.d. \tag{12}$$

According to the Bayesian statistics, we suppose that for each year $t$ when observations were done, mean $m = m(t)$ and variance $v = v(t)$ are random variables having certain distributions. We set a non-informative Jeffrey's prior distribution on these means and variances as we do not have any information about them. Next, using necessary calculations (see the detailed description in Appendix C), we find the posterior distribution of $v$:

$$v \sim InvGamma(\alpha, \beta), \tag{13}$$

where InvGamma is gamma inverse distribution (see Appendix B) and the posterior distribution of $m$ is:

$$m \sim \mathcal{N}(\overline{x}, v/N_p), \tag{14}$$

where the parameters $\alpha = \frac{N_p-1}{2}, \beta = \frac{N_p S}{2}$ and the empirical mean and variance are:

$$\overline{x} = \frac{1}{N_p}\sum_{i=1}^{N_p} x_i, \quad S = \frac{1}{N_p}\sum_{i=1}^{N_p}(x_i - \overline{x})^2. \tag{15}$$

We prove the following lemma about the Bayesian estimates of basal area yearly means $m$ and variances $v$. This statement is a justification for using Bayesian approach for our data.

**Lemma 1.** *Let $x_1(t), x_2(t) \dots, x_{N_p}(t)$—basal area logarithms measured at various forest patches in a certain year $t$; $N_p = N_p(t)$ is the number of patches observed in the year $t$, and $N$ is the total number of patches.*

*(i)    Suppose that for every year $t$ and for every patch, each observation is retained with equal probability $\mathscr{P}$, independently of other observations.*
*(ii)    Assume that for every year $t$, basal area logarithms have finite mean $m(t)$ and variance $v(t)$.*

*Then the Bayesian estimates $m_B(t)$ and $v_B(t)$ of mean and variance $m(t)$ and $v(t)$ approach almost surely to the actual mean $m(t)$ and variance $v(t)$ correspondingly as the total number of patches $N$ goes to infinity: $v_B(t) \xrightarrow{a.s.} v(t)$ and $m_B(t) \xrightarrow{a.s.} m(t)$ as $N \to \infty$.*

**Proof.** To simplify mathematical notation, we omit the year symbol $t$ in the expressions $x_i(t), N_p(t), m(t)$ and $v(t)$, while still implying this time dependence.

By the law of large numbers, $\frac{N_p}{N} \xrightarrow{a.s.} \mathscr{P}$. Hence, $N \to \infty$ implies that $N_p \to \infty$.

Basal area is affected by various random factors. Additionally, all basal area observations are independent of each other and done with equal probability. Hence, we can suppose that the basal area logarithms follow normal distribution with mean $m$ and variance $v$ when the number of patches $N_p$ is big ($N_p \to \infty$):

$$x_1, x_2 \dots, x_{N_p} \sim \mathcal{N}[m, v] \, i.i.d. \tag{16}$$

Let us consider the posterior distribution of the random variable $v$: $v_B \sim InvGamma(\alpha, \beta)$ with $\alpha = \frac{N_p-1}{2}$ and $\beta = \frac{N_p S}{2}$, where $v_B$ is the random variable which we obtain after considering the variance $v$ as a random variable and applying Bayesian theory to it (see Appendix C).

As the random variable $v_B$ has inverse gamma posterior distribution with parameters $\alpha$ and $\beta$, its expectation is:

$$E[v_B|N_p] = \frac{\beta}{\alpha - 1} = \frac{N_p S}{N_p - 3} \to S \text{ as } N_p \to \infty, \tag{17}$$

and the variance

$$Var[v_B|N_p] = \frac{\beta^2}{(\alpha-1)^2(\alpha-2)} = \frac{(N_p S)^2/4}{(N_p-3)^2(N_p-5)/8} \to 0 \text{ as } N_p \to \infty. \tag{18}$$

Next, we consider the mean $m$ as a random variable. After Bayesian update, we obtain that it has normal posterior distribution: $m_B \sim \mathcal{N}\left(\bar{x}, \frac{\nu}{N_p}\right)$, and its expectation is:

$$E[m_B|N_p] = \bar{x} \to m \ \text{ as } \ N_p \to \infty, \tag{19}$$

by the definition (15) of the empirical mean $\bar{x}$.

Regarding the variance of $m_B$,

$$Var[m_B|N_p] = \frac{\nu}{N_p} \to 0 \ \text{ as } \ N_p \to \infty. \tag{20}$$

As $S$ is an empirical variance, it follows that $S \to \nu$ (by definition (15) of the empirical variance); hence, $E[\nu_B|N_p] \to \nu$ as $N_p \to \infty$.

As a result, for the random variables $\nu_B$ and $m_B$ we obtain the following asymptotic:

$$E[\nu_B|N_p] \to \nu \ \text{ and } \ Var[\nu_B|N_p] \to 0 \ \text{ as } \ N_p \to \infty, \tag{21}$$

$$E[m_B|N_p] \to m \ \text{ and } \ Var[m_B|N_p] \to 0 \ \text{ as } \ N_p \to \infty. \tag{22}$$

Now, using (21) and (22), we apply the conditional Chebyshev inequality for the random variables $\nu_B$ and $m_B$:
for any $\epsilon > 0$

$$P(|\nu_B - \nu| \geq \epsilon) \leq \frac{Var[\nu_B|N_p]}{\epsilon^2} \to 0 \ \text{ as } \ N_p \to \infty, \tag{23}$$

and

$$P(|m_B - m| \geq \epsilon) \leq \frac{Var[m_B|N_p]}{\epsilon^2} \to 0 \ \text{ as } \ N_p \to \infty; \tag{24}$$

hence, for any $\epsilon > 0$

$$P(|\nu_B - \nu| \geq \epsilon) \to 0 \ \text{ and } \ P(|m_B - m| \geq \epsilon) \to 0 \ \text{ as } \ N \to \infty. \tag{25}$$

Next, using the almost sure convergence criteria (let $\xi_1, \xi_1$, etc., be random variables; then $\xi_n \xrightarrow{a.s.} \xi \iff \forall \epsilon > 0 : P(\sup_{k \geq n} |\xi_k - \xi| > \epsilon) \to 0$ as $n \to \infty$) and the Equation (25), we obtain that $\nu_B \xrightarrow{a.s.} \nu$ and $m_B \xrightarrow{a.s.} m$ as $N \to \infty$.

This result demonstrates that for any year $t$, the Bayesian estimates of means and variances $m(t)$ and $\nu(t)$ obtained using the posterior distributions (13) and (14), approach the actual means and variances $m(t)$ and $\nu(t)$ correspondingly as the total number of patches $N$ becomes infinitely large. □

These results provide a basis for computer simulations to find Bayesian estimates of the parameters $m$ and $\nu$. For each year $t$, when a certain ecoregion was observed, we simulated the Bayesian posterior estimates of the original parameters $m(t)$ and $\nu(t)$ $N = 1000$ times using to the derived posterior formulas (13) and (14). According to the distributions (13) and (14), we first generate the posterior estimates of $\nu(t)$, and then, based on these results, we obtain the posterior estimates of $m(t)$:

$$\underbrace{\nu(t)}_{\text{prior variance}} \longrightarrow \underbrace{v_1(t), v_2(t), ..., v_N(t)}_{\substack{\text{posterior variances,} \\ \text{N Bayesian simulations}}} \longrightarrow \underbrace{\mu_1(t), \mu_2(t), ..., \mu_N(t)}_{\substack{\text{posterior means,} \\ \text{N Bayesian simulations}}}. \tag{26}$$

Based on posterior estimates of mean and variance, we can obtain certain forest characteristics. Among them is the average basal area growth rate estimate (the average of basal area changes over all years and simulations):

$$\hat{g} = \frac{1}{NT} \sum_{i=1}^{N} [\mu_i(T) - \mu_i(0)]. \tag{27}$$

Consider the partial mean increments $\triangle\mu_i(t) = \mu_i(t) - \mu_i(t-1)$, where $t = \overline{1, T}$ and $i = \overline{1, N}$. Based on these partial mean increments, the standard deviation estimate of basal area is

$$\hat{\sigma}^2 = \frac{1}{NT} \sum_{i=1}^{N} \sum_{t=1}^{T} (\triangle\mu_i(t) - \hat{g})^2. \tag{28}$$

The described approach allows us to evaluate the relative basal area growth rate. This is standard derivation for every ecoregion.

### 2.4. Autoregressive Integrated Moving Average (ARIMA) Models for Basal Area Yearly Averages

In each USA ecoregion, consider the following time series models for $Y_0, Y_1, Y_2, \ldots$ means of basal area logarithms for year $t$: ARIMA (autoregressive integrated moving average) model. First, we define the ARMA (autoregessive moving average) model:

$$Y_t = \underbrace{\alpha_1 Y_{t-1} + \ldots + \alpha_p Y_{t-p}}_{\text{autoregressive (AR) terms}} + \underbrace{\beta_1 \varepsilon_{t-1} + \ldots + \beta_q \varepsilon_{t-q}}_{\text{moving average (MA) terms}} + \varepsilon_t. \tag{29}$$

We denote this model as ARMA($p, q$). In the equation above, $p$ is the order (number of time lags) of the autoregressive (AR) terms, $q$ is the order (number of time lags) of the moving average (MA) terms, $\alpha_i$ are autoregressive parameters, $\beta_i$ are moving average parameters and $\varepsilon_i$ are residual terms: i.i.d. with $\mathbb{E}[\varepsilon_i] = 0$ and $\mathbb{E}[\varepsilon_i^2] < \infty$.

Define the differencing operator as $(\Delta Y)_t := Y_t - Y_{t-1}$. Applied twice, it will look like $(\Delta^2 Y)_t = (Y_t - Y_{t-1}) - (Y_{t-1} - Y_{t-2}) = Y_t + Y_{t-2} - 2Y_{t-1}$. Similarly for $(\Delta^d Y)_t$. If time series $(\Delta^d Y)$ satisfies the autoregressive moving average model ARMA($p, q$), then time series $Y$ satisfies the autoregressive integrated moving average ARIMA($p, d, q$).

In case basal area annual averages $Y_t$ in a certain ecoregion is a non-stationary time series (statistical properties such as mean, variance or autocorrelation of this time series are subject to change over time), we first apply the differencing procedure to the time series $Y_t$ in order to run the ARIMA model (29). Differencing can help to remove non-stationarity of a time series. Consider some common lower order ARIMA models which we will use in the next section. Using the definition (29), we obtain: ARIMA(1,0,0) ($p = 1, d = 0, q = 0$) is autoregressive AR(1) model:

$$Y_t = \alpha_1 Y_{t-1} + \varepsilon_t. \tag{30}$$

ARIMA(0,1,0) ($p = 0, d = 1, q = 0$) is a random walk model: $y_t = \varepsilon_t$, where the difference $y_t = Y_t - Y_{t-1}$; therefore, we obtain a classical random walk formula:

$$Y_t = Y_{t-1} + \varepsilon_t. \tag{31}$$

ARIMA(0,0,1) ($p = 0, d = 0, q = 1$) is moving average MA(1) model:

$$Y_t = \beta_1 \varepsilon_{t-1} + \varepsilon_t. \tag{32}$$

ARIMA(1,1,0) ($p = 1, d = 1, q = 0$) model: $y_t = \alpha_1 y_{t-1} + \varepsilon_t$, where $y_t = Y_t - Y_{t-1}$, and it follows that

$$Y_t = (\alpha_1 + 1)Y_{t-1} - \alpha_1 Y_{t-2} + \varepsilon_t. \tag{33}$$

As we see in the discussion below, either of these four basic ARIMA models can be used to describe dynamics of basal area annual means in the majority of USA ecoregions.

## 3. Results and Discussion

### 3.1. General Statistics

The USA forest inventories data contains information about basal area for various years between 1968 and 2013. However, we do not have this information for every subsequent year in the mentioned interval. Moreover, various forest inventory plots have different numbers of observations, which makes the modeling challenging.

The USA data contains 409,868 observations of 211,949 plots. On average, each plot is observed around two times: $409868/211949 \approx 2$. Among these 211,949 plots: 98,903 were observed once, 61,554 were observed twice, 34,487—three times, 7451—four times, 4125—five times, 4063—six times, 1339—seven times and 27 plots—eight times. For the majority of plots, observations were done with irregular time intervals.

We consider 36 USA ecological regions (see Figure 1). The list of these 36 USA ecological regions is given in Appendix A. Figure 2 contains information about the number of observations for various ecoregions.

Table 1 contains general summary of the considered database. More detailed analysis of dataset reveals that ecoregions have very different number of oservations. For example, ecoregion 262 (California Dry Steppe Province) has only five records, while ecoregion 232 (Outer Coastal Plain Mixed Forest Province) is the most observed one, having 91,856 records. Additionally, different ecoregions had different numbers of years when basal area was observed. This number of years varied from five (in ecoregion 262) to 33 (in ecoregion 232). Within each ecoregion, we have a high correlation between basal area and biomass. This correlation varies from 0.83 (ecoregion 315) to 0.96 (ecoregion 262).

We modeled forest dynamics in the conterminous United States using two forest characteristics: basal area and biomass. These variables are highly correlated in all ecoregions, leading to the very similar time series models. However, estimation of the forest biomass from FIA data involves addition species-specific biomass formulas [35], which have to be parameterized for every tree species and location [13], while forest basal area is computed directly from the data without any additional parameterization [5,13,15]. Therefore, we focused our consideration on the basal area dynamics.

**Table 1.** General statistics of USA data.

| US Results-General | |
|---|---|
| Number of FIA observations totally: | 409 868 |
| 36 ecoregions observed(48 states-all except AK and HI): | 211,212,221,222,223,231,232,234,242, 251,255 |
| | 261,262,263,313,315,321,322,331,332,341,342 |
| | 411,M211,M221,M223,M231,M242,M261,M262 |
| | M313,M331,M332,M333,M334,M341 |
| 40 years:1968-2013 | 1968,1970,1972,1974,1977,1978,1980,1981,1982 |
| | 1983,1984,1985,1986,1987,1988,1989,1990,1991 |
| | 1992,1993,1994,1995,1996,1997,1998,1999,2000 |
| | 2001,2002,2003,2004,2005,2006,2007,2008,2009 |
| | 2010,2011,2012,2013 |

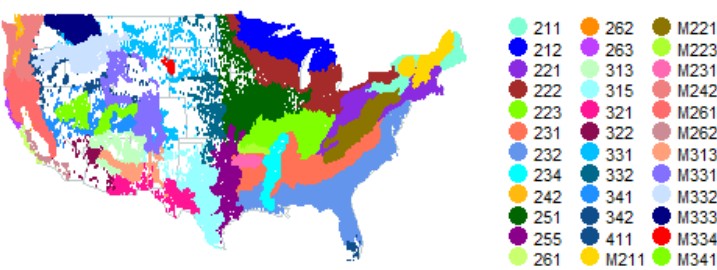

**Figure 1.** USA forest plots belonging to various ecological regions.

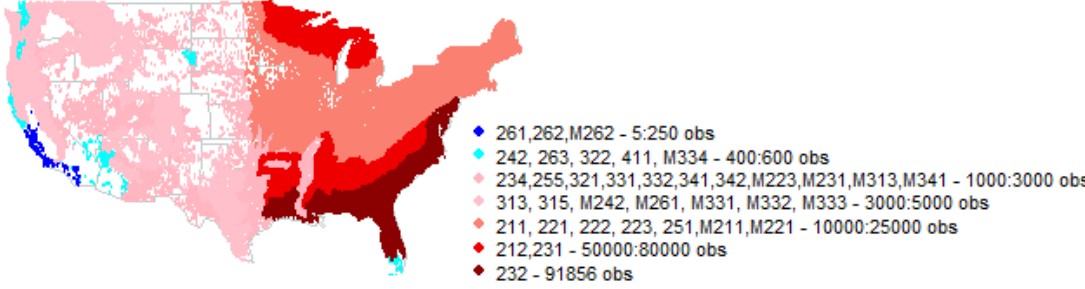

**Figure 2.** Plot observation frequency map. Ecoregions are grouped by the number of observations. For example, 91,856 forest inventory records were done in the ecoregion 232.

### 3.2. Autoregressive Model Ar(1) for Basal Area on Individual Patches

In each USA ecoregion, we built an autoregressive model AR(1) for forest basal area. This model tracks basal area dynamics on every single forest plot:

$$y_p(t) = r + a \cdot y_p(t-1) + \varepsilon_p(t), \tag{34}$$

In the equation above, $p$ is a forest plot, $y_p(t)$ is basal area logarithm at year $t$, $y_p(t-1)$ is the previous year's logarithm of basal area, $r, a$ are model parameters and the random vaiables $\varepsilon_p(t) \sim N(0, \sigma^2)$. On average, each forest plot $p$ was observed only twice, which is not enough to run the model (34) as a classical time series model on a certain patch. Hence, we need to work with the Equation (34) to obtain vectors with more points for model simulations.

As shown in the previous section, from the model (34) the following equation arises:

$$\left[ Ev(a,s) \right]^{-1} \cdot \left[ y(t+s) - a^s \cdot y(t) \right] = \left[ Ev(a,s) \right]^{-1} \cdot Odd(a,s) \cdot r + \delta_p(t,s), \tag{35}$$

where the sums $Ev(a,s) = \left[ 1 + a^2 + \ldots + a^{2(s-1)} \right]^{1/2}$, $Odd(a,s) = \left[ 1 + a + \ldots + a^{s-1} \right]$ and the increments $\delta_p(t,s) \sim \mathcal{N}(0, \sigma^2)$ i.i.d.

The Equation (35) allows us to use all the pairs of basal area observations which were done at the same patch in subsequent years $t$ and $t+s$. Therefore, using the linear regression (35), we employ all existing basal area observations. In the vector Equation (35) (for all existing pairs $t$ and $t+s$), we fix an $a$ value and implement linear regression with respect to $r$. We take $a$ values from the interval $(-2, 2)$ with a step 0.01. For each of these values, we find the model parameter $r$ and the standard error $\sigma$. We look for $a$ values giving the minimal regression standard error (i.e., maximal likelihood function). Based on the simulations results for basal area in all ecoregions (graphs of the regression (35) standard error depending on parameter $a$, QQ plots and normality test results for the model residuals), we conclude the following.

- In ecoregions 211, 212, 221, 222, 223, 231, 232, 234, 251, 255, 313, 321, 322, 331, 332, 341, 342, 411, M211, M221, M223, M231, M313, M331, M332, M333, M334 and M341, the standard error is minimal when parameter $a = 1$, which makes the model (34) for basal area into random walk. However, the residuals for this random walk are not normally distributed according to Shapiro–Wilk normality test.
- In ecoregions 242, M242, M261 and M262 the values $a = \pm 1$ minimize the standard error. For $a = 1$ we have random walk, while the case $a = -1$ is not physical. The residuals for the random walk are not normally distributed.
- In ecoregions 261 and 263 the standard error is minimal when parameter $a = \pm 1$; hence we have random walk. The random walk has normal increments in this case.

- Ecoregions 262 and 315 have a very few observations to model basal area dynamics: in ecoregion 262 we have only five observations, while in ecoregion 315 we do not have paired observations (observations of basal area at the same forest patch in subsequent years).

To summarize, basal area dynamics on individual forest patches can be described as a random walk with Gaussian increments in ecoregions 261 (California Coastal Chaparral Forest and Shrub Province) and 263 (California Coastal Steppe, Mixed Forest and Redwood Forest Province).

In all the remaining USA ecological regions, basal area on individual patches follows a random walk in a logarithmic scale with i.i.d. but with non-normal increments. Hence, in ecoregions distinct from ecoregions 261 and 263, we need to build a better model for basal area to understand its dynamics at the level of individual forest plot.

### 3.3. Autoregressive Model Ar(1) for Basal Area Yearly Averages

### 3.3.1. Frequentist Analysis

Within each ecoregion, we have about 10–30 years during which basal area was observed. For basal area yearly averages we consider a simple autoregressive random walk model:

$$y(t) = y(t-1) + \varepsilon(t), \quad \varepsilon(t) \sim \mathcal{N}(\mu, \rho^2) \quad \text{i.i.d.} \tag{36}$$

Here $y(t)$ is mean of basal area logarithms at the year t; $\mu$, $\rho^2$—model parameters ($\mu$ the average of increments $y(t) - y(t-1)$ and $\rho^2$ is their variance); $\varepsilon(t)$—normally distributed increments.

We implement the model (36) in each USA ecoregion. By looking at the dynamics of basal area yearly averages, QQ plots and Shapiro–Wilk normality test results for the random walk model (36), we obtain the following results:

- Basal area annual means follow a random walk with Gaussian increments in the following ecoregions: 221, 222, 231, 232, 242, 255, 261, 262, 263, 313, 315, 321, 322, 331, 332, 341, 411, M211, M221, M223, M231, M242, M261, M262, M313, M331, M332, M333, M334 and M341;
- In all the rest ecoregions (211, 212, 223, 234, 251 and 342), we need a better model for basal area yearly averages.

As a result, we found that the random walk model (36) describes basal area means in the absolute majority of ecoregions (see Figure 3).

Within each ecoregion, we evaluate the empirical average of basal area growth rate: $[y(T) - y(T_0)] / (T - T_0)$, where $T_0$ is the first year when basal area was observed in an ecoregion, and $T$—the last year (see Figure 4a). From these results, we see that basal area increased mostly in ecoregions M262 (*California Coastal Range Open Woodland Shrub Coniferous Forest Meadow Province*) and 223 (*Central Interior Broadleaf Forest Province*). At the same time, the highest descent was in ecoregions 234 (*Lower Mississippi Riverine Forest Province*), 261 (*California Coastal Chaparral Forest and Shrub Province*) and 263 (*California Coastal Steppe, Mixed Forest and Redwood Forest Province*).

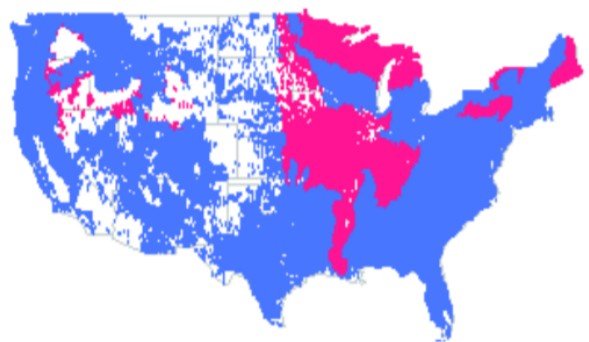

**Figure 3.** Blue areas are inventory plots belonging to ecoregions where basal area means behave as a random walk with normal increments; pink areas—plots in ecoregions where basal area means do not fit the random walk model (36).

The average $\mu$ of the increments $y(t) - y(t-1)$ (differences of basal area averages for the neighbouring years) for all ecoregions was also computed (see Figure 4b). As we see, basal area especially increased in neighbor years in ecoregions 261, 262 (*California Dry Steppe Province*) and M262. The highest decrease was in ecoregions 321 (*Chihuahuan Semidesert Province*) and 313 (*Colorado Plateau Semidesert Province*).

However, these estimates of the average basal area growth rate and the average of neighbor increments are based on strongly uneven observations. Indeed, assume an example wherein we derive the average 5.5 for year 1980 from 1000 observations and the average 4.8 for year 1981 from 10 observations. Then the sampling error for year 1981 is much larger than for year 1980, and the number 4.8 is much less precise than the number 5.5. To compensate for this, we will use Bayesian approach to evaluate forest growth rate accurately.

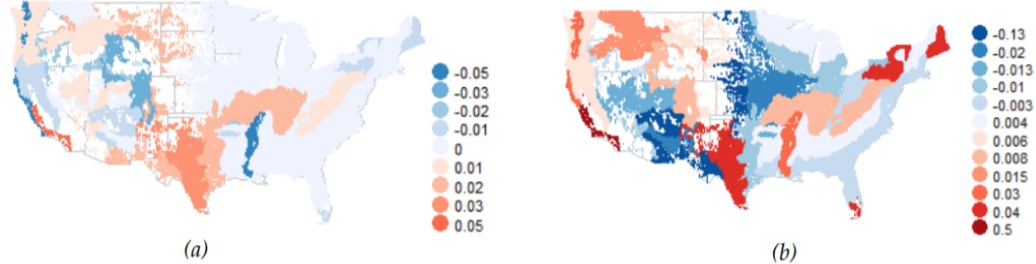

**Figure 4.** (**a**) Empirical average basal area growth rate (over all years when an ecoregion was observed) diagram, and (**b**) empirical means of increments of basal area yearly averages diagram.

3.3.2. Bayesian Analysis

In all 36 USA ecoregions covered in the present work, we have a similar problem as in Quebec: the number of basal area observations is different for different years. For example, in ecoregion 211 (Eastern Broadleaf Forest Province) in the year 1993 we have 1416 observations of basal area, while for in 1997 we have only 20 records. This brought us to idea of using Bayesian approach in order to obtain more reliable estimates of basal area yearly means and variances.

In a Bayesian approach, we put a prior distribution on estimated parameters, and then use the likelihood function from the model to obtain the posterior distribution. Thus, instead of a point estimate for parameters we get a range of values, or, more precisely, a probability posterior distribution for these values. See Appendix C for more technical details.

In a given ecoregion, let $x_1(t), x_2(t) \ldots, x_{N_p}(t)$—basal area logarithms; $N_p$—number of patches observed in year $t$. For all ecoregions, $N_p$ is a big number; hence we can assume that basal area logarithms follow normal distribution:

$$x_1(t), x_2(t) \ldots, x_{N_p}(t) \sim \mathcal{N}\left[m(t), \sigma^2(t)\right] \; i.i.d. \tag{37}$$

Within Bayesian approach, we set a non-informative prior on the parameters $m(t)$ and $v(t)$ ($v(t) = \sigma^2(t)$) in order to find posterior distributions. As a result, we obtain the following posterior distributions of parameters $m(t)$ and $v(t)$:

$$v(t) \sim InvGamma\left(\frac{N_p - 1}{2}, \frac{N_p S}{2}\right), \quad m(t) \mid v(t) \sim \mathcal{N}\left(\overline{x}(t), \frac{v(t)}{N_p}\right), \tag{38}$$

where *InvGamma* is inverse gamma distribution; $\mathcal{N}$—normal distribution—the conditional distribution of $m(t)$ given $v(t)$; $\overline{x}(t)$ and $S(t)$ are empirical mean and variance of basal area at year $t$ (see Formula (15)). Using the posterior distributions (38), we performed computer simulations (26) of basal area means and variances $N = 1000$ times for every year $t$ when a certain ecoregion was observed.

Based on the Bayesian simulations of basal area annual means, we evaluated the average basal area growth rate (forest growth rate) $\hat{g}$ (27) and the standard deviation for yearly fluctuations $\hat{\sigma}^2$ (28) in all ecoregions. Figure 5 is a comparable diagram of Bayesian forest growth rate for all ecoregions.

Based on these results, we have the lowest negative growth rates in the following ecoregions: M261 (*Sierran Steppe Mixed Forest Coniferous Forest Alpine Meadow Province*) (very low), 321 (*Chihuahuan Semidesert Province*) and M313 (*Arizona New Mexico Mountains Semidesert Open Woodland Coniferous Forest Alpine Meadow Province*).

At the same time, the highest positive growth rates are in ecoregions 315 (*Southwest Plateau and Plains Dry Steppe and Shrub Province*) (very high) and 322 (*American Semidesert and Desert Province*).

We obtained the zero growth rate in the ecoregions: 222 (*Midwest Broadleaf Forest Province*), 232 (*Outer Coastal Plain Mixed Forest Province*), 234 (*Lower Mississippi Riverine Forest Province*), 262 (*California Dry Steppe Province*) and 411 (*Everglades Province*).

The obtained Bayesian estimates of forest growth rate are more reliable rather than the frequentist results given above.

For all possible couples of ecoregions involved in the present research, we computed basal area cross-correlations between ecoregions:

$$\hat{\sigma}^2 = \frac{1}{NT} \sum_{i=1}^{N} \sum_{t=1}^{T} (\triangle\mu'_i(t) - \hat{g}')(\triangle\mu''_i(t) - \hat{g}''), \tag{39}$$

where $N = 1000$ is the number of Bayesian simulations, $t = \overline{1, T}$ are all common years when both these ecoregions were observed, $\triangle\mu'_i(t) = \mu'_i(t) - \mu'_i(t-1)$ are the partial increments of simulated means for one ecoregion and $\triangle\mu''_i(t) = \mu''_i(t) - \mu''_i(t-1)$ are partial increments of simulated means for the other ecoregion. Cross-correlation values allow us to see how a rise or drop of basal area in one ecoregion affects basal area dynamics in the other ecoregion. If we have a positive correlation between two ecoregions, then as the value of basal area in one ecoregion rises, so does the value of basal area in another ecoregion. In the case of a negative correlation, the rise of basal area in one ecoregion causes its drop in another ecoregion.

Figure 6 is the diagram of these cross-correlations computed for basal area for all couples of ecoresions. According to these results, we have the highest positive cross-correlation between ecoregions 321 (Chihuahuan Semidesert Province) and M313 (Arizona—New Mexico Mountains Semidesert—Open Woodland—ConiferousForest—Alpine Meadow Province). These two ecoregions are close to each other and share a border. At the same time, the highest negative cross-correlation is between ecoregions 255 (Prairie Parkland (Subtropical) Province—California Coastal Chaparral

Forest and Shrub Province) and 332 (Great Plains Steppe Province). Looking at the ecoregions map, these territories lay close by, having a common point.

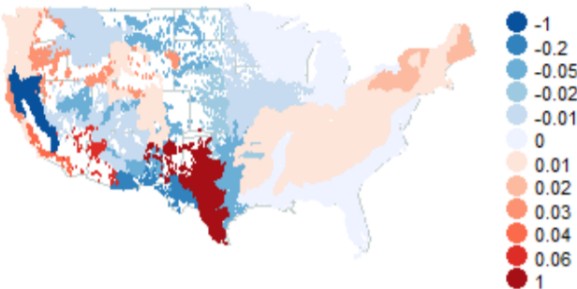

**Figure 5.** Bayesian basal area growth rates ($\hat{g}$) diagram.

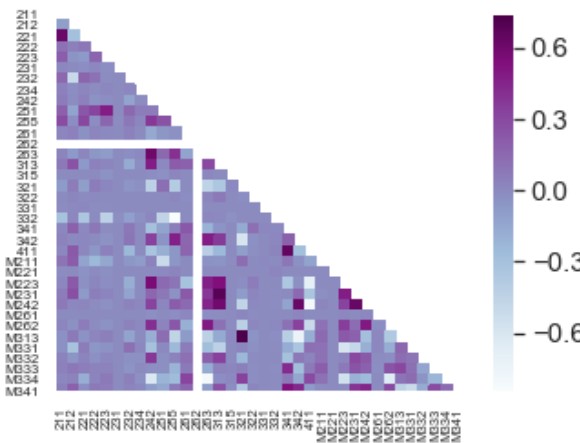

**Figure 6.** Bayesian cross-correlations between USA ecological regions.

### 3.4. Testing Models Using Synthetic Data

We have tested the correctness and efficiency of our time series approach using synthetic data. First, we created $n = 8000$ patches. We generated $m = 60$ basal area values on each of these patch using Laplace distribution (having probability density function $p(x) = 0.5 \cdot a \cdot exp(-a|x|)$ with parameter $a$). The choice of Laplace distribution was based on the fact that we wanted to model forest inventory observations which have large errors. Laplace distribution has fatter tails compared to the normal distributionp hence we preferred the Laplace distribution for our synthetic data. We chose $n$ and $m$ according to the existing forest inventory data dimensions for an average sized ecoregion. Next, for every patch, we removed every $k = 5$th basal area value among $m$ generated values. Removing these basal area values is similar to what we have in the original forest data: basal area was observed with uneven year gaps. This way, we left $k/m = 5/60 \approx 8\%$ of the data which we simulated initially using Laplace distribution.

We ran our AR(1) models for the synthetic data, and obtained that synthetic basal area values follow the same pattern: at a single patch level, basal area values behave as a random walk with non-normal tails, while basal area annual averages follow Gaussian random walk.

### 3.5. ARIMA (Autoregressive Integrated Moving Average) Models for Basal Area Annual Averages

In every USA ecoregion, we implemented ARIMA models (29) of various orders $(p, d, q)$ for basal area annual averages. We chose the "best" fitting model with the least possible number of parameters in every ecoregion. As a result, based on model residuals, we obtained that the following ARIMA models better describe basal area annual means in the corresponding ecoregions:

ARIMA(1,0,0) (AR(1) model (30))—ecoregions 263, 313, 315, 331, 342, 411 and M223.

ARIMA(0,1,0) (random walk (31))—ecoregions 232, 242, 341, M221, M261 and M262.

ARIMA(0,0,1) (MA(1) model (32))—ecoregions 231, 251, 321, 332, M231, M242 and M341.

ARIMA(1,1,0) model (33)—ecoregions 322, M332, M333 and M334.

The results above are visualized in Figure 7. In the other ecoregions, ARIMA models of higher orders can be used to describe basal area dynamics. However, we prefer to use the models of lower orders as they involve less parameters. As an example, illustrations of particular ARIMA models are given at Figures A1 and A2.

From all these ARIMA results, the following information follows. First, in the mountain territories (M332, M333 and M334) of Dry USA ecological domain, ARIMA(1,1,0) model fits the best basal area annual means among all the other ARIMA models. From Equation (33) of ARIMA(1,1,0) model, we conclude that in these mountain territories the present year basal area values are strongly dependent on the previous two years basal area values. Looking at ARIMA(1,0,0) (AR(1)) model, we see that it is the best fit for the ecoregions which also belong to Dry Domain. Regarding the random walk ARIMA(0,1,0) model and ARIMA(0,0,1) (MA(1) model), they are the best mainly for ecoregions from Humid USA Domain.

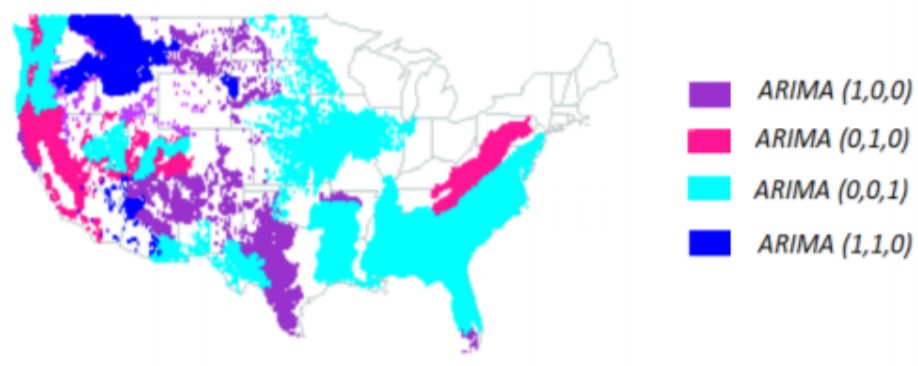

**Figure 7.** ARIMA models for the basal area yearly averages map.

*3.6. General Discussion*

Autoregressive Modeling of Forest Dynamics

In the present paper, we have applied ideas from financial engineering to model forest basal area dynamics. First, we considered the similarity between basal area and a stock market index similarly to how we did in [16]. For this, we used autoregressive AR(1) model to describe basal area dynamics. We obtained that in the majority of USA ecoregions, basal area behaves similarly to a stock market. However, in certain territories the similarity was not detected; the forest and the stock market are obviously different systems even though some similarity certainly exists.

Regarding autoregressive AR(1) model, in the absolute majority of USA ecoregions, the results are similar to our previous analysis of forest dynamics in Quebec [16]. Specifically, basal area is extremely volatile (highly swings around the mean value) with non-Gaussian oscillations, while basal area annual averages follow geometric random walk with normal tails. This is analogous to dynamics of individual stock and a stock market correspondingly. In detail, for almost all USA ecoregions, we have the AR(1) model for basal area on individual patches with parameter $a = 1$, which makes AR(1) process a random walk. However, this random walk has heavy tails. Since AR(1) process converges to its stationary distribution only for $-1 < a < 1$, we obtain that basal walks away in both positive and negative directions and does not converge in the long run. We have a slightly different picture in California Coastal Province (ecoregions 261 and 263), where basal area on individual forest plots perfectly fits random walk with normal increments.

In this work, we justified the use of Bayesian techniques from quantitative finance for USA forest inventory data in order to treat uneven observations. This Bayesian approach allowed us to obtain a map of comparable forest growth rate which reveals the areas with the most rapid and slow forest growth dynamics.

Finally, in all considered USA ecoregions, we generalized our research of autoregressive models by implementing various time series ARIMA models for basal area annual averages. We detected ARIMA(1,1,0) model is the best for areas with high elevation situated in Dry Domain. At the same time, the ecoregions where we choose ARIMA(0,1,0) or ARIMA(0,0,1) models belong mainly to the Humid Domain. This all leads to the conclusion that the present year basal area values are strongly linked with the previous years values in the territories mainly belonging to Dry Domain.

*3.7. Challenges, Limitations and Future Research*

In order to apply the classic patch-mosaic methodology [36,37], a forest considers a large landscape unit as a patch mosaic system, where all patches evolve according to the same macroscopic that patch dynamics can be described in terms of various macroscopic variables changing in time. The trees on a single forest patch are influenced by a variety of diverse natural and anthropogenic factors [38,39]. However, different forest plots within the same ecoregion can be exposed to different environmental conditions depending on the local topography, proximity to water reservoirs and altitude. In particular, mountain ecoregions present a special challenge due to the very large variation of climatic and disturbance conditions and water balance, which can change substantially with altitude and slope expositions. In addition, there could an uncertainty in the definition of the ecoregion boundaries. Moreover, the ecoregion classifications could be quite different depending on the selection of primary environmental factors and statistical methodology [33,34,40]. For example, we based our work on the Bailey's classification of the US ecoregions developed by the US Forest Service [33,34]; however, there is another classification of the US ecoregions developed by EPA [40]. There are statistical approaches that we would like to employ in future studies to reduce this ambiguity. In particular, we can select forest patches within the ecoregion that are similar by climatic conditions and forest tolerances [41] and soil moisture and successional patthways [15,42].

Since for the most ecoregions basal area on individual forest plots cannot be described as random walk with normal increments, it is reasonable to concentrate future research on individual plots and use use heavy-tailed increments. We can check autocorrelation for individual plots and use, if necessary, autoregressive models. One can use Pareto-type or asymmetric Laplace distributions. Since there is a lot of missing data, we will have to account for this, either in Bayesian or frequentist framework. Finally, we can study the correlation between an individual patch and the entire ecoregion, analogously to beta in financial theory. We can also try to fill missing data using techniques similar to Kalman filtering.

Due to computational efficiency and analytical tractability, the developed time series models can be particularly useful for the large scale biogeochemical modeling and global climate change modeling [17], where the species-level community dynamics are usually not taken into account or averaged. These time series models can be also linked to the large scale models of forest tolerance in order to quantify environmental disturbance regimes and forecast overall basal area and biomass dynamics in resilience modeling [41].

## 4. Conclusions

We have conducted a time series analysis of forest dynamics in the conterminous USA using USDA FIA dataset. Basal area yearly averages behave as geometric a random walk with normal increments in most of the ecoregions. In California Coastal Province, geometric random walk with normal increments adequately describes the dynamics of both basal area yearly averages and basal area on individual forest plots. In all other ecoregions, basal areas of individual forest patches behave as random walks with heavy tails. We have also implemented time series ARIMA models for annual

averages basal area in every USA ecological region. The developed models account for stochastic effects of environmental disturbances and allow one to forecast forest dynamics. Time-series analysis can be employed for forecasting of forest dynamics at the large scale for the global climate change and biogeochemistry modeling.

**Author Contributions:** Conceptualization, A.S. and N.S.; methodology, A.S. and O.R.; software, O.R.; statistical analysis, O.R.; writing—original draft preparation, O.R.; writing—review and editing, A.S. and N.S.; visualization, O.R.; supervision, A.S. and N.S.; project administration, A.S. and N.S. All authors have read and agreed to the published version of the manuscript.

**Funding:** This work was partially supported by a grant from the Simons Foundation (283770 to N.S.).

**Conflicts of Interest:** The authors declare no conflict of interest.

## Abbreviations

The following abbreviations are used in this manuscript:

| | |
|---|---|
| QQ | quantile-quantile (for a plot) |
| AR | autoregressive process |
| AR(1) | autoregressive process of order 1 |
| ARIMA | autoregressive integrated moving average model |
| USDA | United States Department of Agriculture |
| FIA | USDA Forest Service Forest Inventory and Analysis Program |
| GIS | Geographic Information Systems |
| i.i.d. | independent and identically distributed (random variables) |
| $\xrightarrow{a.s.}$ | almost sure convergence |

## Appendix A. Usa Ecological Subdivisions

The following division of USA territory took place: Domains → Divisions → Provinces.

Below is the list of ecological regions covered in the present paper (where the first digit identifies the domain (2—Humid Domain, 3—Dry Domain and 4—Humid Tropical Domain):

- 211: Northeastern Mixed Forest Province
- 212: Laurentian Mixed Forest Province
- 221: Eastern Broadleaf Forest Province
- 222: Midwest Broadleaf Forest Province
- 223: Central Interior Broadleaf Forest Province
- 231: Southeastern Mixed Forest Province
- 232: Outer Coastal Plain Mixed Forest Province
- 234: Lower Mississippi Riverine Forest Province
- 242: Pacific Lowland Mixed Forest Province
- 251: Prairie Parkland (Temperate) Province
- 255: Prairie Parkland (Subtropical) Province
- 261: California Coastal Chaparral Forest and Shrub Province
- 262: California Dry Steppe Province
- 263: California Coastal Steppe, Mixed Forest and Redwood Forest Province
- 313: Colorado Plateau Semidesert Province
- 315: Southwest Plateau and Plains Dry Steppe and Shrub Province
- 321: Chihuahuan Semidesert Province
- 322: American Semidesert and Desert Province
- 331: Great Plains Palouse Dry Steppe Province
- 332: Great Plains Steppe Province
- 341: Intermountain semidesert and Desert Province

- 342: Intermountain semidesert Province
- 411: Everglades Province
- M211: Adirondack New England Mixed Forest and Coniferous Forest, Alpine Meadow Province
- M221: Central Appalachian Broadleaf Forest Coniferous Forest Meadow Province
- M223: Ozark Broadleaf Forest Meadow Province
- M231: Ouachita Mixed Forest Meadow Province
- M242: Cascade Mixed Forest and Coniferous Forest Alpine Meadow Province
- M261: Sierran Steppe Mixed Forest and Coniferous Forest Alpine Meadow Province
- M262: California Coastal Range Open Woodland and Shrub Coniferous Forest Meadow Province
- M313: Arizona-New Mexico Mountains Semidesert and Open Woodland Coniferous Forest Alpine Meadow Province
- M331: Southern Rocky Mountain Steppe and Open Woodland Coniferous Forest Alpine Meadow Province
- M332: Middle Rocky Mountain Steppe and Coniferous Forest Alpine Meadow Province
- M333: Northern Rocky Mountain Forest and Steppe Coniferous Forest Alpine Meadow Province
- M334: Black Hills Coniferous Forest Province
- M341: Nevada-Utah Mountains Semidesert and Coniferous Forest Alpine Meadow Province

## Appendix B. Gamma and Gamma Inverse Distributions

Gamma distribution with shape $\alpha$ and scale $\beta$ has the following probability density function:

$$\text{density} \quad f(x; \alpha, \beta) = \frac{\beta^\alpha}{\Gamma(\alpha)} x^{\alpha-1} e^{-\beta x}, \quad x > 0. \tag{A1}$$

The following relationship between gamma and gamma inverse distributions takes place: a random variable $X$ has gamma inverse distribution with parameters $\alpha$ and $\beta$ if a random variable $\frac{1}{X}$ has Gamma distribution with parameters $\alpha$ and $\frac{1}{\beta}$:

$$X \sim InvGamma(\alpha, \beta) \quad if \quad \frac{1}{X} \sim Gamma\left(\alpha, \frac{1}{\beta}\right). \tag{A2}$$

## Appendix C. Posterior Distributions Calculation

In the present section, we derive the posterior distributions of basal area means $m$ and variances $v$ using techniques from Bayesian statistics. Means and variances are the random variables with certain distributions. We choose a non-informative Jeffrey's prior distribution on $m$ and $v$: $\pi(m, v) \propto v^{-1}$. Then, using a series of calculations, we will obtain the posterior distributions of the means and variances based on the chosen prior distribution.

Let basal area logarithms $x_1, x_2 \ldots, x_{N_p}$, where $N_p$ is the number of patches observed, follow normal distribution with mean $m$ and variance $v$:

$$x_1, x_2 \ldots, x_{N_p} \sim \mathcal{N}[m, v] \, i.i.d. \tag{A3}$$

Consider the likelihood function of normal distribution $l(x_i | m, v) = \frac{1}{\sqrt{2\pi v}} exp\left[-\frac{(x_i - m)^2}{2v}\right]$. Based on it, the cumulative likelihood function is:

$$L(x_1, ..., x_n | m, v) = (2\pi v)^{-\frac{n}{2}} \cdot exp\left[-\frac{nS}{2v}\right] \cdot exp\left[-\frac{n(m - \overline{x})^2}{2v}\right], \tag{A4}$$

where $\bar{x}$ and $S$ are the empirical mean and variance:

$$\bar{x} = \frac{1}{N_p} \sum_{i=1}^{N_p} x_i, \quad S = \frac{1}{N_p} \sum_{i=1}^{N_p} (x_i - \bar{x})^2. \tag{A5}$$

Posterior distributions are calculated according to Bayesian formula for posterior distribution:

$$f(\xi|X) = \frac{f(\xi) \cdot f(X|\xi)}{\int f(\xi) f(X|\xi) \, d\xi}, \tag{A6}$$

where $\xi$ is parameters vector, $X$—random vector.

Using the formula (A6) and setting a non-informative prior on the parameters $m$ and $v$ $\pi(m,v) \propto v^{-1}$, we obtain:

$$p(m,v|x_1,...,x_n) = \frac{\frac{1}{v}(2\pi v)^{-\frac{n}{2}} \cdot exp\left[-\frac{nS}{2v}\right] \cdot exp\left[-\frac{n(m-\bar{x})^2}{2v}\right]}{\int \int \frac{1}{v}(2\pi v)^{-\frac{n}{2}} \cdot exp\left[-\frac{nS}{2v}\right] \cdot exp\left[-\frac{n(m-\bar{x})^2}{2v}\right] dm dv}, \tag{A7}$$

from which we derive the posterior distribution of $m$ and $v$:

$$p(m,v|x_1,...,x_n) = \frac{1}{\Gamma(\alpha)} \cdot \sqrt{\frac{N_p}{2\pi}} \cdot v^{-n/2-1} \cdot exp[-\beta/v] \cdot \beta^\alpha \cdot exp\left[\frac{-(m-\bar{x})^2}{2v}\right], \tag{A8}$$

where the parameters $\alpha = \frac{N_p-1}{2}$, $\beta = \frac{N_p S}{2}$ and $\Gamma(\alpha)$ is Euler's Gamma function: $\Gamma(\alpha) = \int_0^\infty \frac{t^{\alpha-1}}{e^t} dt$.

Now, based on the posterior distribution (A8), we can find the posterior distribution of the variance $v$:

$$p(v|x_1,...,x_n) = \int_{-\infty}^{\infty} p(m,v|x_1,...,x_n) dm = v^{-2} \cdot \frac{\beta^\alpha}{\Gamma(\alpha)} \left(\frac{1}{v}\right)^{\alpha-1} \cdot exp[-\beta/v], \tag{A9}$$

where $\frac{\beta^\alpha}{\Gamma(\alpha)} \left(\frac{1}{v}\right)^{\alpha-1} \cdot exp[-\beta/v] = Gamma(\frac{1}{v}; \alpha, \beta)$—gamma distribution function, from which we obtain the posterior distribution with $\alpha = \frac{N_p-1}{2}$ and $\beta = \frac{N_p S}{2}$:

$$v \sim InvGamma(\alpha, \beta) = InvGamma\left(\frac{N_p-1}{2}, \frac{N_p S}{2}\right). \tag{A10}$$

From (A8), we also derive the posterior distribution of $m$:

$$m \mid v \sim \mathcal{N}(\bar{x}, v/N_p). \tag{A11}$$

## Appendix D. ARIMA Models Results

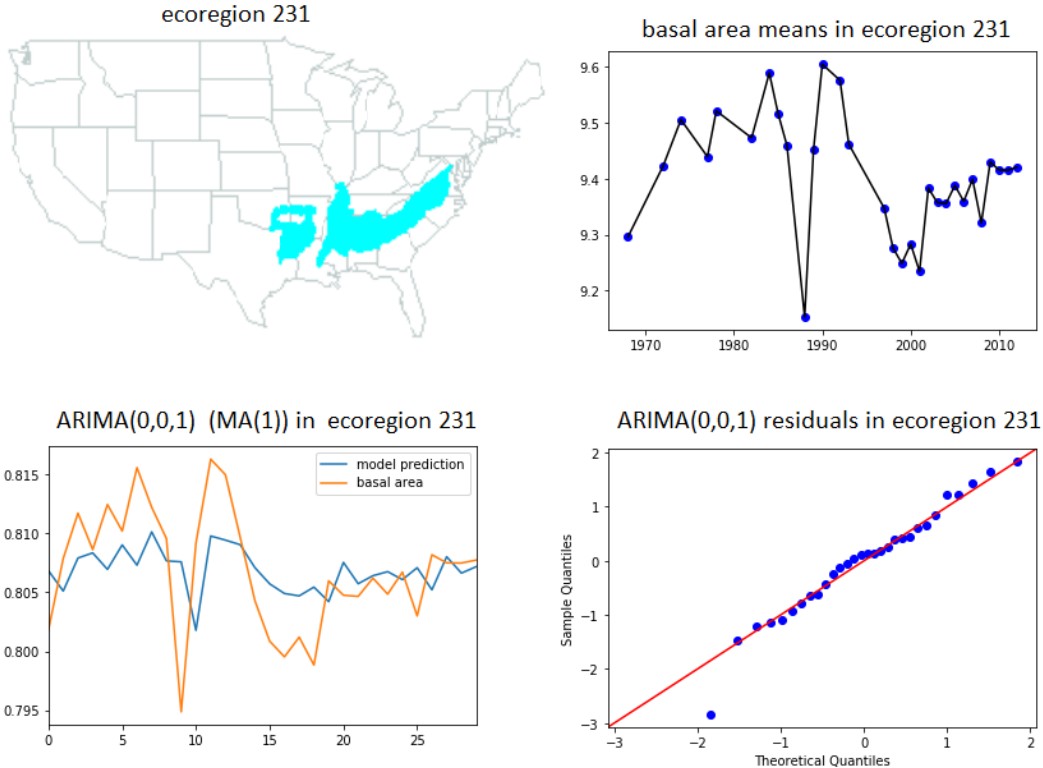

**Figure A1.** ARIMA (0,0,1) (MA(1)) model results for basal area annual means in ecoregion 231 (Southeastern Mixed Forest Province).

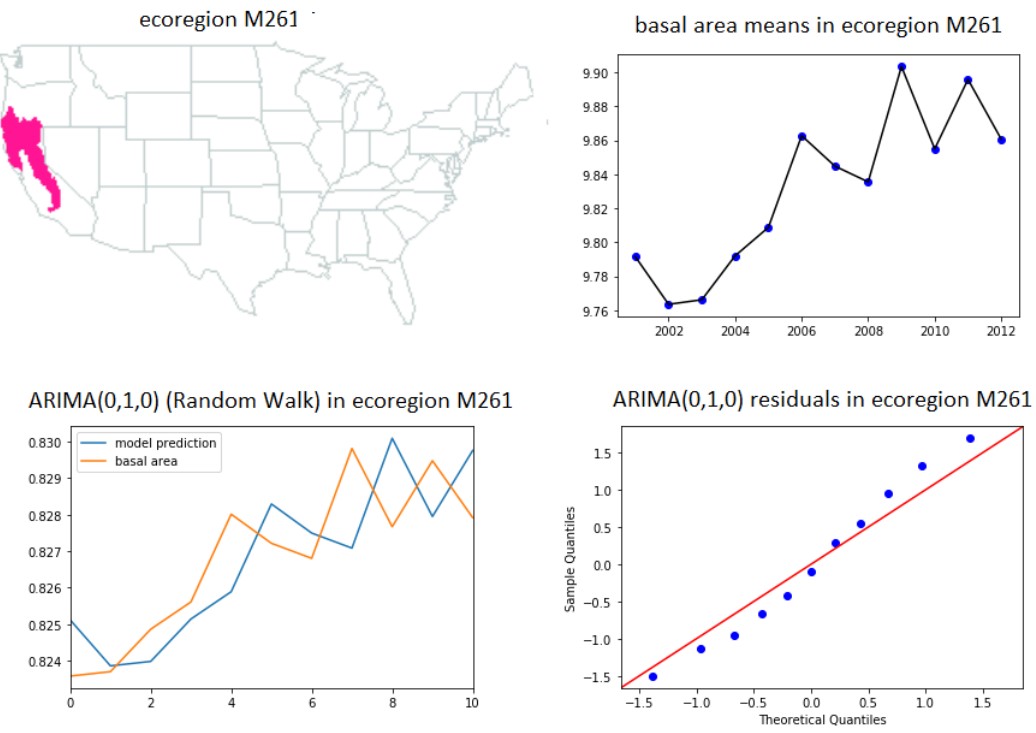

**Figure A2.** ARIMA (0,1,0) model results for basal area annual means in ecoregion M261 (Sierran Steppe Mixed Forest Coniferous Forest).

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
