# Peer review of "Time Series Analysis of Forest Dynamics at the Ecoregion Level"

_forecasting, doi:10.3390/forecast2030020_

Round 1

Reviewer 1 Report

The authors attempted in their paper to model forest dynamics at the ecoregion spatial scale level by using time series ARIMA models on basal area data, which forms an important forest management variable.  The authors used a huge amount of data derived from the USDA Forest Inventory data set for different ecoregions of the U.S.A. and their methodological approach is sound and valid.  The paper is very well structured, written, and presented and I think it provides useful insights for forest and environmental management at this spatial scale level.  I recommend publication of the paper in Forecasting pending only the following minor revisions:

  • The state-of-the-art section on forest dynamics modeling should be expanded and additional references should be added. Almost 40% of the references cited in the article are self-citations.
  • The following spelling errors need to be corrected:

Line 138.  “We” replaced by “we”

Line 337.  “masal” replaced by “basal”

Reviewer 2 Report

The paper proposes a time series analysis approach for forest growth in US ecoregions. Forest growth dynamics is captured by an autoregressive model used at the ecoregion level. Four main instances of the model are identified, which fit different US ecoregions data. The modelling techniques come from approaches previously applied in the stock market domain. The proposed model has been previosly applied to Quebec forest growth data.

The paper is insteresting, technically sound and clear. The proposed approach is for sure valuable for the modeling of forest growth dynamics and results obtained on US data are interesting. Therefore, my overall opinion is that the paper deserves publication.

I had some problems in following some of the more technical parts of the paper. This is due to the fact that the authors assume that the reader is familiar with some of the advanced mathematical and statistical tools they use. This is fine... the average reader will probably skip some technical details and focus on the modelling method and results. However, a couple of aspects should be clarified in order to allow the reader to grasp some important aspects of the approach.

In particular, in the definition of the ARIMA(p,d,q) model (page 8), the author should clarify how the differencing procedure is involved in the model formula. At the moment this is explained informally, but this does not make it easy for the reader to understand the procedure. As a consequence, it is difficult to understand, for instance, how equation (33) is obtained. Can you give a general formalization of  ARIMA(p,d,q) by giving a formula that uses also d?

Moreover, several times (e.g., at the end of page 12) the authors state that the Bayesian approach for the computation of averages is necessary since the different number of data items in different years make standard averages unreliable. This is probably perfectly clear to an experienced reader, but I think it would be better to clarify this point in order to let the average reader understand why standard averages are unreliable and why the Bayesian approach is better.
